# Pharmacodynamic Interactions between Puerarin and Metformin in Type-2 Diabetic Rats

**DOI:** 10.3390/molecules27217197

**Published:** 2022-10-24

**Authors:** Zhen Li, Shengguang Wang, Xinyu Wang, Peng Gao, Shiming Zhang, Yingning Mo, Dongsheng Zhao, Long Dai

**Affiliations:** 1School of Pharmacy, Binzhou Medical University, Yantai 264003, China; 2School of Pharmacy, Shandong University of Traditional Chinese Medicine, Jinan 250355, China; 3School of Traditional Chinese Medicine, Shandong University of Traditional Chinese Medicine, Jinan 250355, China; 4Pharmaceutical Research Institute, Shandong University of Traditional Chinese Medicine, Jinan 250355, China; 5School of Management, Shandong University of Traditional Chinese Medicine, Jinan 250355, China

**Keywords:** puerarin, metformin, type-2 diabetes, antihyperglycaemia, antihyperlipedaemia, anti-inflammatory

## Abstract

Herb–drug interactions are vital in effectively managing type-2-diabetes complications. Puerarin is a natural isoflavonoid in the Pueraria genus, and its pharmacological activities, including antidiabetic activity, are well established. The similar modes of action of puerarin and metformin in diabetic models suggest their positive pharmacodynamic interactions. This study investigated this in streptozotocin/nicotinamide-induced type-2 diabetic rats. Puerarin at doses of 80 mg/kg, 120 mg/kg and 160 mg/kg improved the activity of metformin in reversing hyperglycaemia, dysregulated lipid profiles, dysfunction of the liver, kidney, and pancreas, and inflammation. The treatment with either puerarin (high dose, 160 mg/kg intraperitoneally) or metformin (100 mg/kg intraperitoneally) did not bring the dysregulated biomarkers to normal levels in 4 weeks. By contrast, the combination of puerarin (160 mg/kg) and metformin (100 mg/kg) did. This study is the first to report scientific evidence for the positive pharmacodynamic interactions between puerarin and metformin.

## 1. Introduction

Blood-glucose levels are elevated in patients with diabetes, a chronic metabolic disease [1]. Uncontrolled blood glucose levels for prolonged periods cause damage to the various body organs (kidney, eyes, blood vessels, nerves, heart, etc.) [2]. Diabetes is categorised into three types; type-1, type-2, and gestational diabetes. Type-2 diabetes, also known as diabetes mellitus, is predominant in adults and accounts for 90% of all diabetes cases [3]. Diabetes occurs due to either decreased insulin production by the pancreas or the body’s inability to use insulin effectively (insulin resistance) [4]. The occurrence of diabetes has increased alarmingly over the years. About 537 million adults live with diabetes, which was responsible for 6.7 million deaths in 16 July 2021 (https://diabetesatlas.org/ accessed on 1 September 2022). It is predicted that the number of adults with diabetes will rise to 643 million in 2030 and 783 million by 2045. Interestingly, 3 in 4 adults with diabetes live in low- and middle-income countries.

Numerous reviews highlight medicinal plants’ value [4,5] and their interactions with antidiabetic drugs in the clinic [6,7]. In traditional Chinese medicine, plants belonging to the genus Pueraria (*P. lobata* [8], *P. tuberosa* [9], *P. mirifica* [10], *P. thunbergiana* [11], and *P. montana* [12]) are being used to treat inflammation [13], diabetes [14], cardiovascular diseases [14], cerebrovascular diseases [15], etc. In all these plants, puerarin (isoflavone glycoside) is the major bioactive constituent. Iminosugars mimic monosaccharide sugars and inhibit glycosidase enzyme; thus, they have received increased attention as antidiabetic agents [16,17,18,19,20]. A small number of scientific investigations [21,22,23] were carried out to elucidate the antidiabetic activity and mechanisms of puerarin. Puerarin is reported to exhibit antihyperglycaemic activity by recovering the physiological homeostasis functions of the liver, pancreas, adipose tissue, and skeletal muscle [24]. It activates the glucagon-like peptide 1 (GLP-1R) signalling pathway in the pancreas of diabetic rats. The activation of the GLP-1R pathway resulted in (1) the inhibition of apoptosis via the upregulation of Akt and downregulation of Caspase-3, and (2) the promotion of β-cell renewal via the upregulation of the Wnt/β-catenin and JAK2/STAT3 pathways [25,26]. It also improved the mitochondria’s function via the upregulation of oxidative phosphorylation and the detoxification of reactive oxygen species (ROS) [27]. In the skeletal and adipose tissues of diabetic rats, puerarin activates glucose transporter 4 (GLUT4) transmission via the upregulation of serum β-endorphin, protein kinase B (Akt), and cannabinoid type 1 receptor (Cb1) [28,29]. The activation of GLUT4 promotes glucose transport [30]. In the livers of diabetic rats, puerarin activated the phosphoinositide 3-kinase (PI3K)/Akt signalling pathway, which subsequently upregulated the phosphorylation of Foxo1 and the inhibition of Glucose-6-phosphatase (G6pase) and phosphoenolpyruvate carboxykinase (PEPCK) [31,32]. 

Numerous medicinal plants were reported to enhance the activity of metformin in diabetic animal models [33]. Protopanaxadiol ginsenoside from *Panax ginseng* showed improvements in plasma glucose and insulin levels when co-administered with metformin [34]. *Momordica charantia* extract showed a significant improvement in the antihyperglycaemic effect of metformin upon concurrent administration [35]. The representative examples of plants that improved the antihyperglycaemic effect of metformin are prickly pear cactus pad [36], *Trigonellafoenum*
*gracum* [37], and *Allium sativum* [38]. 

There are no studies on the possible pharmacodynamic interactions between puerarin and metformin. Thus, this study evaluated the pharmacodynamic activities of the concurrent administration of puerarin and metformin on hyperglycaemia, hyperlipidaemia, oxidative stress, inflammation, and pancreas architecture in streptozotocin-nicotinamide-induced diabetes in Sprague–Dawley (SD) rats.

## 2. Results

Type-2 diabetes was successfully induced in the SD rats by injecting streptozotocin/nicotinamide (intraperitoneal (IP) route). Metformin (100 mg/kg, IP route) was used as a standard drug. The dose-dependent effects of puerarin (80, 120, and 160 mg/kg) on reversing hyperglycaemia, dysregulated lipidaemia, oxidative stress, and proinflammatory markers were analysed. The pharmacodynamic interactions between the puerarin and the metformin were analysed. The duration of the intervention was four weeks. In the following sections, the statistically significant results are discussed.

### 2.1. Effects of Pharmacodynamic Interactions between Puerarin and Metformin on Hyperglycaemia

Upon the administration of streptozotocin/nicotinamide to the SD rats, the blood-glucose levels were raised to 190.84 ± 1.50 mg/dL from 75.35 ± 2.13 mg/dL. The puerarin showed a dose-dependent effect and reversed the elevated blood-glucose levels at 120 mg/kg and 160 mg/kg, whereas it did not reverse the glucose levels at 80 mg/kg. The glucose levels in the rats treated with the high puerarin dose (160 mg/kg) were 136.03 ± 1.24 mg/dL, while the levels in the metformin-treated rats were 126.76 ± 1.05 mg/dL. Neither the puerarin nor the metformin reduced the glucose levels to those observed in the normal rats. The co-administration of the puerarin (at all doses) potentiated the effect of the metformin in reducing the elevated blood glucose levels. The combination of puerarin and metformin (160 mg/kg and 100 mg/kg, respectively) reduced the elevated glucose levels (80.43 ± 2.20 mg/dL) to normal. These effects are illustrated in Figure 1A.

### 2.2. Effects of Pharmacodynamic Interactions between Puerarin and Metformin on Dysregulated Lipidaemia

The streptozotocin/nicotinamide increased the cholesterol (from 81.89 ± 0.48 mg/dL to 119.60 ± 0.65), triglycerides (from 67.82 ± 0.36 mg/dL to 133.60 ± 0.53 mg/dL), and low-density lipoprotein (LDL) from 14.14 ± 0.69 mg/dL to 35.33 ± 0.26 mg/dL, and reduced the high-density lipoprotein (HDL) from 46.08 ± 1.00 mg/dL to 23.98 ± 0.16 mg/dL. The puerarin regulated the dysregulated lipidaemia at doses of 120 mg/kg and 160 mg/kg; however, the levels did not reach normal values. The cholesterol, triglycerides, and LDL and HDL levels in the rats treated with puerarin (160 mg/kg) were 102.20 ± 1.07 mg/dL, 107.70 ± 0.82 mg/dL, 26.96 ± 0.63 mg/dL, and 36.00 ± 0.35 mg/dL, respectively. These levels in the metformin group were 96.51 ± 0.40 mg/dL, 94.83 ± 0.51 mg/dL, 23.85 ± 0.37 mg/dL, and 39.07 ± 0.33 mg/dL, respectively; these values are significantly different from those observed in normal rats. The puerarin (at all doses, the except 80 mg/kg dose on HDL) potentiated the metformin’s effects in regulating the lipid profile. The co-administration of puerarin (160 mg/kg) and metformin (100 mg/kg) reversed the dysregulated lipid profile to the normal level (cholesterol, 83.47 ± 0.46 mg/dL; triglycerides, 69.18 ± 0.76 mg/dL; LDL, 15.15 ± 0.40 mg/dL; and HDL, 48.79 ± 1.35 mg/dL). The effects on cholesterol (Figure 1B), triglycerides (Figure 1C), HDL (Figure 1D), and LDL (Figure 1E) are illustrated in Figure 1.

### 2.3. Effects of Pharmacodynamic Interactions between Puerarin and Metformin on Kidney Function

The effects of puerarin and its interaction with metformin on kidney function in the type-2 diabetic rats were assessed by analysing the serum-urea and creatinine levels. The intraperitoneal administration of streptozotocin/nicotinamide elevated the serum-urea (34.54 ± 0.37 mg/dL to 90.17 ± 0.41 mg/dL) and creatinine (0.34 ± 0.03 mg/dL to 1.58 ± 0.03 mg/dL) levels. The intervention with puerarin showed a dose-dependent effect in reducing the elevated levels; however, it did not show a significant effect at a low dose (80 mg/kg). The serum-urea and creatinine levels in the puerarin (high dose, 160 mg/kg)-treated group were 73.09 ± 0.73 mg/dL and 1.12 ± 0.03 mg/dL, respectively; these values are still higher than those observed in the normal rats. The metformin (100 mg/kg, IP route) also did not reduce these values (urea, 47.93 ± 0.81 mg/dL; creatinine, 0.80 ± 0.01 mg/dL) to normal. The co-administration of puerarin (at all doses) and metformin showed a greater effect than either metformin or puerarin alone in reducing the elevated urea and creatinine levels. The combination of a high dose of puerarin (160 mg/kg) and metformin (100 mg/kg) did reduce the levels (35.61 ± 0.30 mg/dL of urea, 0.31 ± 0.01 mg/dL) to normal. The effects on the serum urea are described in Figure 1F and on creatinine in Figure 1G.

### 2.4. Effects of Pharmacodynamic Interactions between Puerarin and Metformin on Liver Function

The influence of puerarin, alone or in combination with metformin, was assessed by measuring the liver-enzyme (alanine transaminase (ALT), aspartate transaminase (AST), and alkaline phosphatase (ALP)) levels. The serum ALT, AST, and ALP levels in the normal rats (27.74 ± 0.45 IU/L, 69.67 ± 0.68 IU/L, and 63.51 ± 0.60 IU/L, respectively) were elevated in the diabetic rats (73.25 ± 0.49 IU/L, 127.50 ± 0.93 IU/L, and 135.70 ± 0.69 IU/L, respectively). The puerarin (except at a low dose, 80 mg/kg) showed a dose-dependent effect in reducing the elevated ALT, AST, and ALP levels, and at a high dose (160 mg/kg), the levels were 62.12 ± 0.35 IU/L, 105.30 ± 0.72 IU/L, and 115.00 ± 0.47 IU/L, respectively; these were still higher than the normal values. The ALT, AST, and ALP levels in the metformin-treatment group were 51.53 ± 0.60 IU/L, 92.15 ± 0.29 IU/L, and 91.52 ± 0.47 IU/L, respectively; these values were still far from the normal values. The puerarin, at all doses, significantly potentiated the bioactivity of the metformin in restoring the liver function and brought the values (ALT, 29.54 ± 0.47 IU/L; AST, 72.34 ± 0.39 IU/L; and ALP, 64.96 ± 0.58 IU/L) back to normal at its high dose (160 mg/kg), together with the metformin. The effects on theALT are shown in Figure 1H, on the AST in Figure 1I, and on te ALP in Figure 1J.

### 2.5. Effects of Pharmacodynamic Interactions between Puerarin and Metformin on Oxidative Stress

The effects of the puerarin and its pharmacodynamic interaction with the metformin on the oxidative stress induced by the streptozotocin/nicotinamide in the rats were evaluated by analysing the glutathione (GSH), malondialdehyde (MDA), and total antioxidant capacity (TAC) levels in the serum. In the diabetic rats, the GSH levels were raised to 2.47 ± 0.02 nmol/μL (2.02 ± 0.03 nmol/μL in the normal rats) and the MDA levels were raised to 6.43 ± 0.06 nmol/μL (2.68 ± 0.02 nmol/μL in the normal rats), and the TAC levels were reduced to 238.70 ± 1.06 nmol/μL (359.50 ± 1.39 nmol/μL in the normal rats). The elevated levels of GSH and MDA were reduced by the puerarin (120 mg/kg; GSH, 2.37 ± 0.01 nmol/μL; MDA, 5.79 ± 0.01 nmol/μL and 160 mg/kg; GSH, 2.29 ± 0.01 nmol/μL; MDA, 5.19 ± 0.02 nmol/μL only) and metformin (100 mg/kg; GSH, 2.23 ± 0.01 nmol/μL; and MDA, 4.67 ± 0.03 nmol/μL); however, the levels were still high compared to those in the normal rats. The puerarin at 120 mg/kg and 160 mg/kg increased the TAC levels to 270.06 ± 0.63 nmol/μL and 285.70 ± 1.44 nmol/μL, whereas the metformin increased the TAC levels to 300.60 ± 0.58 nmol/μL; these values were lower than those in the normal control. The puerarin at all doses increased the activity of the metformin. In the group treated with a combination of high-dose puerarin and metformin, the values (GSH, 2.08 ± 0.01 nmol/μL; MDA, 2.89 ± 0.04 nmol/μL; and TAC, 354.10 ± 2.55 nmol/μL) were equivalent to those found in the normal rats. Figure 2A–C describes the effects on the GSH, MDA, and TAC.

### 2.6. Effects of Pharmacodynamic Interactions between Puerarin and Metformin on Inflammation

The levels of proinflammatory cytokines (interleukin (IL)-1β, IL-6, and tumour-necrosis factor alpha (TNF-α)) were measured to assess the activity of puerarin and its pharmacodynamic interaction with metformin. The levels of proinflammatory cytokines were increased in the streptozotocin/nicotinamide-induced diabetic rats. The IL-1β, IL-6, and TNF-α levels in the normal rats were 80.91 ± 0.39 pg/mL, 25.45 ± 0.18 pg/mL, and 15.17 ± 0.38 pg/mL, respectively, whereas these levels in the diabetic rats were 260.70 ± 0.21 pg/mL, 61.29 ± 0.38 pg/mL, and 50.95 ± 0.16 pg/mL, respectively. The puerarin at a low dose did not show an effect on the elevated proinflammatory cytokines, while at medium and high doses, it showed a dose-dependent effect in reversing the elevated proinflammatory cytokines. At its highest dose (160 mg/kg), the levels were 163.80 ± 0.43 pg/mL (IL-1β), 49.51 ± 0.40 pg/mL (IL-6), and 36.84 ± 0.45 pg/mL (TNF-α); these values were significantly higher than those observed in the normal control rats. In the metformin group, the levels of IL-1β were 120.90 ± 0.26 pg/mL, the levels of IL-6 were 41.55 ± 0.37 pg/mL, and the levels of TNF-α were 30.24 ± 0.46 pg/mL. The co-administration of the puerarin (at all doses) improved the activity of the metformin in reversing the elevated proinflammatory cytokines. The combination (puerarin, 160 mg/kg + metformin, 100 mg/kg) reduced the levels (IL-1β, 81.86 ± 0.37 pg/mL; IL-6, 26.85 ± 0.30 pg/mL; and TNF-α, 15.41 ± 0.41 pg/mL) to those in the normal rats. These results are described in Figure 2D (IL-1β), Figure 2E (IL-6), and Figure 2F (TNF-α).

### 2.7. Histology Studies

The antidiabetic effects of the puerarin, the metformin, and their combination were further investigated by analysing the histological changes in the islets’ size and number in pancreatic tissue. The streptozotocin/nicotinamide reduced the size and number of islets. The puerarin (at 120 mg/kg, medium dose; 160 mg/kg, high dose) and metformin (100 mg/kg) slightly increased the islet numbers. The size and number of the islets were increased in the rats that were given puerarin and metformin together. The representative photomicrographs are shown in Figure 3. The insulin secretion is associated with islets’ size and number. Thus, in this case, increased islet number and size also contributed to the increased activity of the metformin in the presence of the puerarin.

## 3. Discussion

Herbal medicines are plants or parts that can protect health or treat diseases [39,40]. Globally, the use of herbal medicines is continuously growing, especially for managing chronic diseases such as diabetes, dyslipidaemia, hypertension, cancer, neurological diseases, inflammatory diseases, etc. [41,42,43,44,45,46]. The main reasons for the increasing popularity of herbal medicines are historical, cultural, and psychosocial [47]. In many countries, it is reported that people with diabetes use herbal medicines in addition to modern medicine [48,49,50]. Herb–drug interactions are reported to exert contradictory health effects: the positive side is improved therapeutic activity [51,52], and the negative side is the occurrence of adverse events [53,54]. Some studies have reported the beneficial effects of herb–drug interactions in diabetes management [7]. The beneficial effects can be either additive or synergistic through pharmacodynamic or pharmacokinetic interactions. 

Puerarin is one a bioactive isoflavone that is predominantly found in kudzu plants, such as Radix Puerarin, *Pueraria lobata* [8], *Pueraria tuberosa* [9], *Pueraria mirifica* [10] *Pueraria thunbergiana* [11] Pueraria montana [12], etc. Puerarin is traditionally known to alleviate inflammation [13] and is used to treat diabetes [14], cardiovascular [14] and cerebrovascular [15] diseases, etc. Recently, research studies have attempted to elucidate the mechanism of puerarin in type-2 diabetes and its related complications [21,22,23]. Various in vivo studies of puerarin described the hypoglycemic mechanism to work through the liver, pancreas, adipose tissue, and skeletal muscle [24]. In the pancreas, puerarin has been shown to enhance the glucagon-like peptide 1 (GLP-1R) signalling pathway. The signal transduction of GLP-1R results in the inhibition of apoptosis via the upregulation of Akt and the downregulation of Caspase-3, as well as in the promotion of β-cell renewal via the upregulation of Wnt/β-catenin and Janus kinase 2 (JAK2)/Signal transducer and activator of transcription 3 (STAT3) pathways [25,26]. It was also reported that puerarin improved the function of mitochondria through the upregulation of oxidative phosphorylation and detoxifying reactive oxygen species (ROS) [27]. In skeletal and adipose tissue, puerarin activates the transmission of the glucose transporter 4 (GLUT4) via the upregulation of serum β-endorphin, Akt, and Cb1 [28,29]. The activation of GLUT4 subsequently promotes glucose transport [30]. In the liver, puerarin was found to activate the PI3K/Akt signalling pathway, which subsequently upregulates the phosphorylation of Foxo1 and the inhibition of Glucose-6-phosphatase (G6pase) and phosphoenolpyruvate carboxykinase (PEPCK) [31,32]. 

Metformin is a first-line, safe, and inexpensive oral antidiabetic drug that is used to manage the complications of type-2 diabetes [55]. The major mode of action of metformin is the decrease in hepatic gluconeogenesis. Its beneficial actions are mediated through the liver, small intestines, and skeletal muscle. In the liver, it (1) decreases gluconeogenesis by activating the serine/threonine kinase 11-5′adenosine monophosphate (AMP)-activated kinase (AMPK) signalling pathway and inhibiting mitochondrial-electron transport and adenyl cyclase, (2) reduces insulin resistance by reducing selenoprotein P levels, thus activating AMPK, and (3) inhibits mitochondrial glycerol phosphate dehydrogenase, thus inhibiting the synthesis of glucose from glycerol. In the small intestine, it (1) increases plasma Glucagon-like peptide 1 (GLP-1), thus inhibiting gluconeogenesis, and (2) activates the AMP kinase. In the skeletal muscle, it (1) increases the translocation of GlUT-4 to the plasma membranes to increase the glucose uptake and (2) activates AMPK α2 to enhance glucose disposal. 

From the reported studies, it is clear that there are few similarities in the mechanisms of action between metformin and puerarin. Thus, we hypothesised that there would be positive pharmacodynamic interactions between metformin and puerarin in diabetes models. The present study tested this hypothesis in a streptozotocin/nicotinamide-induced type-2 diabetes rat model. Based on the literature cited in the introduction, puerarin was administered intraperitoneally to diabetic rats at 80 mg/kg, 120 mg/kg, and 160 mg/kg. The metformin dose was fixed at 100 mg/kg, based on the literature cited in the introduction. In this study, the duration of the treatment was set as 4 weeks to determine whether there is a positive pharmacodynamic interaction between metformin and puerarin. The literature reported that puerarin and metformin require several weeks to reverse diabetes complications completely. 

The classical diagnostic marker of type-2 diabetes is blood hyperglycaemia, and uncontrolled hyperglycaemia leads to a dysregulated lipid profile, inflammation, and dysfunction of the liver, kidney, and pancreas. Thus, in this study, we analysed the glucose levels, lipid profile (cholesterol, triglycerides, LDL and HDL), kidney-function markers (urea and creatinine), liver-function markers (ALT, AST and ALP), proinflammatory cytokines (IL-1β, IL-6, and TNF-α), and pancreas function (islets’ size and number). The puerarin did not show a significant effect at a low dose (80 mg/kg), and the results agree with those reported in the literature. However, the puerarin did show a dose-dependent effect at 120 mg/kg and 160 mg/kg in reversing dysregulated glucose and lipid profiles, kidney, liver, and pancreas function, and inflammation. However, the puerarin did not completely reverse the dysregulated levels to normal, even at a high dose (160 mg/kg), during the 4-week treatment period. The same scenario was also observed with the metformin treatment.

In comparison, the concurrent administration of puerarin (at all doses, of 80 mg/kg, 120 mg/kg, and 160 mg/kg) improved the bioactivity of the metformin, which was significantly higher than with the treatment with either puerarin or metformin alone. All the dysregulated biomarkers in the diabetic rats reached the normal level through the combination of puerarin (160 mg/kg) and metformin (100 mg/kg) in 4 weeks. 

The improved antidiabetic activity of the combination could also have been due to pharmacokinetic interactions between the puerarin and the metformin. Puerarin is reported to inhibit CYP450 enzymes (CYP3A4, CYP2B6, and CYP2C9) and p-glycoprotein [56,57,58]; thus, it affects the pharmacokinetics of herbs and drugs [59,60,61,62,63,64]. However, there are no reports on the pharmacokinetic interactions between metformin and puerarin. Metformin is mainly metabolised by CYP2C11, CYP2D1, and CYP3A1/2 [65]. Thus, from the reported studies, it is understood that different CYP isoforms are involved in the metabolism of metformin and puerarin; thus, it is improbable to expect CYP-mediated pharmacokinetic interactions between them. A small number of studies have reported the p-glycoprotein inhibitory activity of metformin [66,67]. In addition, both metformin and puerarin inhibit the p-glycoprotein through a similar mechanism of inhibiting the activation of NF-κB and cAMP-responsive element-binding protein (CREB) [67,68]. Thus, there might not be p-glycoprotein-mediated pharmacokinetic interactions between puerarin and metformin. However, other mechanisms might evoke pharmacokinetic interactions between them; thus, it is worth investigating these pharmacokinetic interactions to obtain insights into how to encourage pharmacodynamic interactions between puerarin and metformin. Ours is the first study to report the antidiabetic effect of the puerarin-plus-metformin combination. By contrast, the majority of the reports in the literature are on plant extracts containing puerarin. In this study, the compounds were given intraperitoneally; however, the oral route is preferred. In addition, intraperitoneal administration does not allow the pharmacokinetic interactions to be investigated. Thus, further studies are warranted to (1) confirm the beneficial pharmacodynamic interactions via the oral route, (2) investigate the pharmacokinetics and (3) examine the relationship between pharmacodynamics and pharmacokinetics. 

## 4. Materials and Methods

Puerarin, streptozotocin, nicotinamide, and citrate buffer were purchased from Shanghai Yuanye Biotechnology Co., Ltd. (Shanghai, China). Metformin was purchased from Youcare Pharmaceutical Group Co., Ltd. Male Sprague–Dawley rats were purchased from Vital River (Weitong Lihua) Laboratory Animal Technology Co., Ltd. (Beijing, China). The rodent chow diet was purchased from Keao Xieli Feed Co., Ltd. (Beijing, China). The required glassware was purchased from Sichuan Shubo (Group) Co., Ltd. (Chongzhou, China). The polypropylene consumables were purchased from Axygen Biotechnology (Hangzhou) Co., Ltd. (Hangzhou, China). The automatic biochemical analyser (URIT8026, URIT Medical Electronics, Guizhou, China) was used to measure the serum biochemical parameters. The ELISA kits used to measure oxidative stress markers (glutathione (GSH), malonaldehyde, and total antioxidant capacity) were purchased from Jiangsu Meimian industrial Co., Ltd. (Jiangsu, China) and proinflammatory markers (IL-1β, IL-6, and TNF-α) were purchased from Thermo Fisher Scientific (China) Co., Ltd. (Shanghai, China). FilterMax™ F-3-Multi-Mode Microplate Reader (Molecular Devices, LLC. San Jose, CA, USA) was used to measure the absorbance/fluorescence of the test solutions. The histology of the pancreas was observed using a microscope slide scanner PANNORAMIC MIDI II (3DHISTECH Ltd., Budapest, Hungary).

### 4.1. Experimental Design

Type-2 diabetes was induced using streptozotocin (STZ) and nicotinamide by following the procedure in the literature [69], with a few modifications. The Animal Ethics Committee of Shandong University of Traditional Chinese Medicine (SDUTCM20211015002) approved the experimental protocol. The male SD rats (175 ± 28 g, 8–10 weeks old) were housed in an animal house (maintained at 25 ± 3 °C, 55 ± 5% humidity, with a 12-hour light–dark cycle) for a week before the start of the experiment. Five rats were kept in each cage and given free access to a standard rodent-chow diet and water. All the rats were weighed and randomly divided into control and experimental groups, each consisting of 10 rats. On experiment day 1, the experimental rats fasted for 8 h and were provided with water before the induction of diabetes. Nicotinamide was dissolved in 0.9% sodium chloride to prepare a solution of 230 mg/mL and injected intraperitoneally (IP, dose: 230 mg/kg or 1 mL/kg) 15 min before streptozotocin administration. The streptozotocin solution (32.5 mg/mL) was freshly (immediately before injection) prepared by dissolving it in 50 mM of sodium-citrate buffer (pH 4.5) and administered intraperitoneally (dose: 64 mg/kg or 2.0 mL/kg). The control animals received citrate buffer intravenously (dose: 2 mL/kg). The rats continued to receive normal food and drinking water. On experiment day 10, the blood-glucose levels (tail-vein blood sample) were measured using a One Touch Basic blood-glucose-monitoring system. The rats with blood-glucose levels greater than 150 mg/dL (8.3 mmol/L) were considered diabetic. The treatment protocol (shown in Table 1) was developed based on those reported in the literature [33].

The effect of puerarin on diabetic rats was investigated at three doses (80 mg/kg, 120 mg/kg, and 160 mg/kg). The doses were selected based on those reported in the literature [25,27,70,71,72,73,74,75,76,77,78,79,80,81]. The intraperitoneal dose of metformin was fixed at 100 mg/kg based on values reported in the literature [82,83]. Puerarin and metformin were dissolved in phosphate-buffered saline (PBS, pH 7.4), and the required volumes were administered intraperitoneally once a day for four weeks. After the treatment, the rats were sacrificed by cervical dislocation under diethyl-ether anaesthesia. The blood was collected by cardiac puncture for biochemical analysis, and the pancreas were excised for histological analysis. The sera were separated from the blood by centrifugation. Biochemical parameters (glucose, cholesterol, triglycerides, high-density lipoprotein (HDL), low-density lipoprotein (LDL), urea, creatinine, alkaline phosphatase (ALP), aspartate transaminase (AST), alanine transaminase (ALT), glutathione (GSH), malondialdehyde (MDA), and total antioxidant capacity (TAC)) were measured. A small portion of pancreatic tissue was used for histology studies. All the tissues and sera were stored at −80 °C immediately after collection for further experiments.

### 4.2. Biochemical Experiments

The serum biochemical parameters of glucose, cholesterol, triglycerides, HDL, LDL, urea, creatinine, ALP, AST, and ALT were measured using an automatic biochemical analyser (URIT8026, URIT Medical Electronic, Shenzhen, China), following the manufacturer’s instructions. The levels of oxidative-stress markers (GSH, MDA, and TAC) and proinflammatory cytokines (IL-1β, IL-6, and TNF-α) were measured using respective ELISA kits following the manufacturer’s instructions.

### 4.3. Histology Studies

The pancreas were immediately (after dissection) perfused with 10% aqueous formaldehyde solution and then embedded in paraffin. Tissue sections of 5 µm were prepared using a microtome and stained with haematoxylin and eosin (H & E). The stained sections were analysed using a microscope slide scanner. The histologies of the pancreas were qualitatively assessed by counting the number and sizes of islets.

### 4.4. Statistical Analysis

GraphPad Prism (Version 9.4.1 for Windows, GraphPad Software, San Diego, CA, USA) was used to perform the statistical analysis. Results are presented as mean ± standard error of the mean (SEM) of a minimum of ten readings. The statistical significance of the effects of interventions was calculated using one-way analysis of variance (ANOVA) followed by Tukey’s multiple-comparisons test. The data were considered significant at *p* < 0.05.

## Figures and Tables

**Figure 1 molecules-27-07197-f001:**
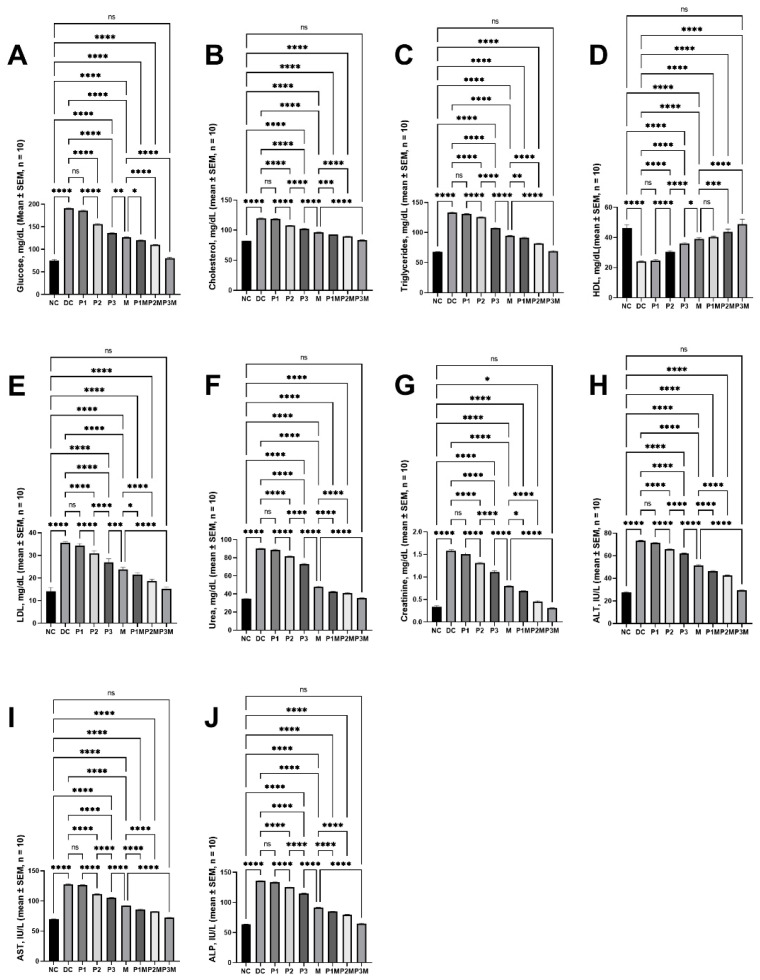
The effects of the pharmacodynamic activity of puerarin and metformin and the pharmacodynamic interaction between puerarin and metformin on serum biochemical parameters. (**A**) Hyperglyceamia, (**B**) cholesterol, (**C**) triglycerides, (**D**) HDL, (**E**) LDL, (**F**) urea, (**G**) creatinine, (**H**) ALT, (**I**) AST, (**J**) ALP. The groups are represented as follows: NC, normal control; DC, diabetic control; P1, puerarin (80 mg/kg); P2, puerarin (120 mg/kg); P3, puerarin (160 mg/kg); M, metformin (100 mg/kg); P1M, puerarin (80 mg/kg) + metformin (100 mg/kg); P2M, puerarin (120 mg/kg) + metformin (100 mg/kg); P3M, puerarin (160 mg/kg + metformin (100 mg/kg).The annotations on the pairwise comparisons are as follows: ns, not significant (*p* > 0.05); *, *p* ≤ 0.05; **, *p* ≤ 0.01; ***, *p* ≤ 0.001; ****, *p* ≤ 0.0001.

**Figure 2 molecules-27-07197-f002:**
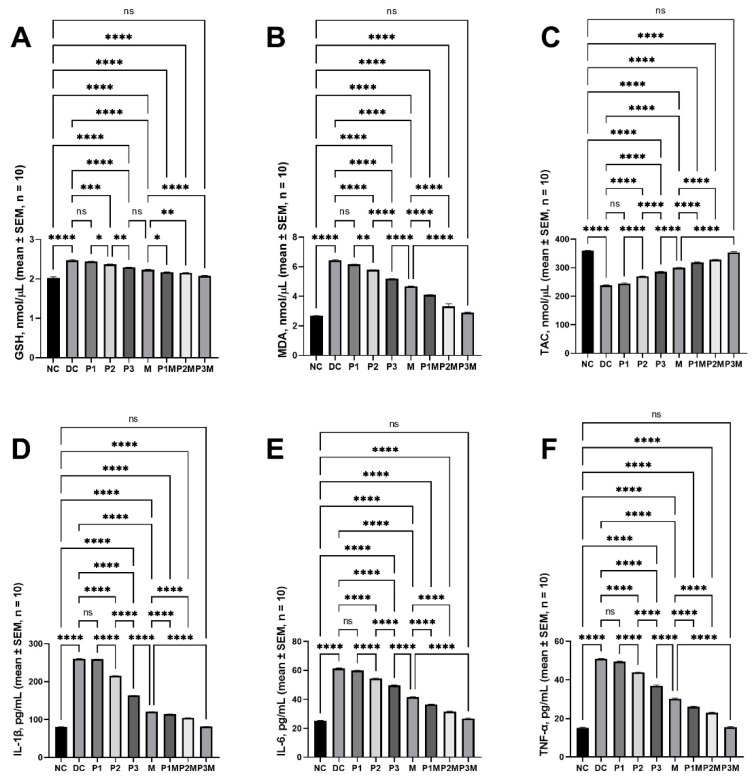
The pharmacodynamic activity of puerarin and metformin and the effects of the pharmacodynamic interaction between puerarin and metformin on the oxidative stress and inflammation. (**A**) GSH, (**B**) MDA, (**C**) TAC, (**D**) IL-1β, (**E**) IL-6, and (**F**) TNF-α. The groups are represented as follows: NC, normal control; DC, diabetic control; P1, puerarin (80 mg/kg); P2, puerarin (120 mg/kg); P3, puerarin (160 mg/kg); M, metformin (100 mg/kg); P1M, puerarin (80 mg/kg) + metformin (100 mg/kg); P2M, puerarin (120 mg/kg + metformin (100 mg/kg); P3M, puerarin (160 mg/kg) + metformin (100 mg/kg). The annotations on the pairwise comparisons are as follows: ns, not significant (*p* > 0.05); *, *p* ≤ 0.05; **, *p* ≤ 0.01; ***, *p* ≤ 0.001; ****, *p* ≤ 0.0001.

**Figure 3 molecules-27-07197-f003:**
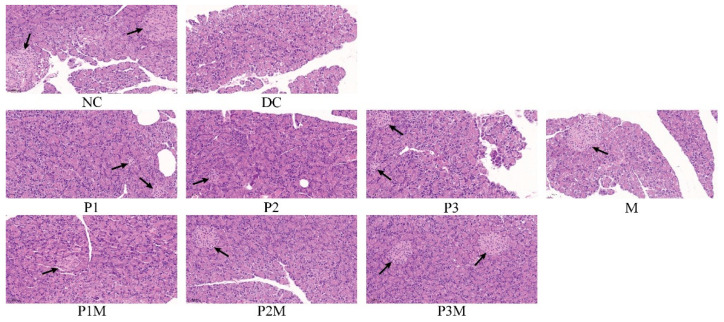
The histology of the pancreas. The black arrows show the locations of islets in the pancreas. The groups are represented as follows: NC, normal control; DC, diabetic control; P1, puerarin (80 mg/kg); P2, puerarin (120 mg/kg); P3, puerarin (160 mg/kg); M, metformin (100 mg/kg); P1M, puerarin (80 mg/kg) + metformin (100 mg/kg); P2M, puerarin (120 mg/kg) + metformin (100 mg/kg); P3M, puerarin (160 mg/kg) + metformin (100 mg/kg).

**Table 1 molecules-27-07197-t001:** Grouping of rats and intervention.

Group	Number of Rats	Intervention	Dose, Administration Route
NC	10	PBS	2 mL/kg, IP
DC	10	PBS	2 mL/kg, IP
P1	10	Puerarin	80 mg/kg, IP
P2	10	Puerarin	120 mg/kg, IP
P3	10	Puerarin	160 mg/kg, IP
M	10	Metformin	100 mg/kg, IP
P1M	10	Puerarin + Metformin	80 mg/kg, IP + 100 mg/kg IP
P2M	10	Puerarin + Metformin	120 mg/kg, IP + 100 mg/kg IP
P3M	10	Puerarin + Metformin	160 mg/kg, IP + 100 mg/kg IP

Note: IP, intraperitoneal; PBS, phosphate-buffered saline.

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
