# Peer review of "Pharmacodynamic Interactions between Puerarin and Metformin in Type-2 Diabetic Rats"

_molecules, 2022, doi:10.3390/molecules27217197_

Round 1

Reviewer 1 Report

In this manuscript the authors studied “Pharmacodynamic interactions between puerarin and metformin in type2 diabetic rats” by combination of fix doses of both. They have studied all possible adverse physiological effects like hyperglycemia, dysregulated lipid profile, dysfunction of liver, kidney, and pancreas, inflammation etc. and found that combination of these two drugs have shown better results than the individual administration. This is a kind of drug-herb interaction and can be better therapeutic model in future.

My comments are as follows:

1. The title of the manuscript is presented correctly; abstract is written flawlessly and represents the manuscript rationally.

(The structure of metformin and puerarin may be presented in this section)

2. The experiments for Pharmacodynamic interactions between puerarin and metformin on hyperglycaemia/on dysregulated lipidaemia/on kidney function/on liver function/on oxidative stress/on inflammation are performed with authentic methods.

a) The standard dose for puerarin are 80, 120 and 160 mg/Kg and for metformin is 100 mg/Kg. What is the rationale behind this?

b) The representational figures 1 and 2 are not properly visible, the resolution should be enhanced.

c) In all cases co-administration of puerarin (160 mg/Kg) and metformin (100 mg/Kg) reversed to the normal conditions, and this seems to be a good result.

d) Is there any study on metformin dose higher than 100 mg/Kg and puerarin 160 mg/Kg (separately).?

e) The histology studies clearly showed the enhancements of the sizes and numbers of islets of pancreatic beta-cells upon administration of combined dose, and it is the positive side of this work.

f) The mechanistic studies of the drug-drug interaction (if-any) may be provided in this work.

3. The authors are recommended to emphasis the importance of iminosugars and sugar derivatives as an anti-diabetic agents, and it is recommended to cite following relevant articles related to iminosugars in introduction section.

  1. Nash, R. J.; Kato, A.; Yu, C-. Y.; Fleet, G. W. J. Iminosugars as therapeutic agents: recent advances and promising trends. Future Med. Chem. 2011, 3, 1513−1521.
  2. Yang, L.-F.; Shimadate, Y.; Kato, A.; Li, Y.-X.; Jia, Y.-M.; Fleet, G.W.J.; Yu, C.-Y. Synthesis and glycosidase inhibition of N-substituted derivatives of DIM. Org. Biomol. Chem. 2020, 18, 999–1011.
  3. Chennaiah, A.; Dahiya, A.; Dubbu, S.; Vankar, Y. D. A Stereoselective Synthesis of an Imino Glycal: Application in the Synthesis of (−)-1-Epi -Adenophorine and a Homoiminosugar. Eur. J. Org. Chem. 2018, 6574−6581.
  1. Chennaiah, A.; Bhowmick, S.; Vankar, Y. D. Conversion of glycals into vicinal-1,2-diazides and 1,2-(or 2,1)-azidoacetates using hypervalent iodine reagents and Me3SiN3. Application in the synthesis of N-glycopeptides, pseudo-trisaccharides and an iminosugar. RSC Adv. 2017, 7, 41755−41762.
  1. Rajasekaran, P.; Ande, C.; Vankar, Y. D. Synthesis of (5,6 & 6,6)-oxa-oxa annulated sugars as glycosidase inhibitors from 2-formyl galactal using iodocyclization as a key step. ARKIVOC 2022, vi, 5−23.

Overall, after addressing the points mentioned above, I recommend this article to publish in molecules.

Author Response

In this manuscript the authors studied “Pharmacodynamic interactions between puerarin and metformin in type2 diabetic rats” by combination of fix doses of both. They have studied all possible adverse physiological effects like hyperglycemia, dysregulated lipid profile, dysfunction of liver, kidney, and pancreas, inflammation etc. and found that combination of these two drugs have shown better results than the individual administration. This is a kind of drug-herb interaction and can be better therapeutic model in future.

My comments are as follows:

  1. The title of the manuscript is presented correctly; abstract is written flawlessly and represents the manuscript rationally.

(The structure of metformin and puerarin may be presented in this section)

Response: Thanks for the suggestion. However, incorporating these two structures as a figure will increase the number of figures in the manuscript, and we have to revise all the figures’ numbers. In addition, the chemical structures of metformin and puerarin are well known. So, we did not include the chemical structures. I hope this will be aggregable to you.

  1. The experimentsfor Pharmacodynamic interactions between puerarin and metformin on hyperglycaemia/on dysregulated lipidaemia/on kidney function/on liver function/on oxidative stress/on inflammation are performed with authentic methods.
  2. a) The standard dose for puerarin are 80, 120 and 160 mg/Kgand for metformin is 100 mg/KgWhat is the rationale behind this?

Response: Based on the literature, we have selected the doses of puerarin and metformin.

  1. b) The representational figures 1and 2are not properly visible, the resolution should be enhanced.

Response: The resolution of the figures is enhanced.

  1. c) In all cases co-administration of puerarin (160 mg/Kg) and metformin (100 mg/Kg) reversed to the normal conditions, and this seems to be a good result.

Response: Yes, the combination works well.

  1. d) Is there any study on metformin dose higher than 100 mg/Kgand puerarin 160 mg/Kg(separately).?

Response: Yes, there were a few studies in which the metformin dose in rats was more than 100 mg/Kg, and the puerarin dose was 160 mg/Kg. We have selected the doses based on the majority of the publications.

  1. e) The histology studies clearly showed the enhancements of the sizes and numbers of islets of pancreatic beta-cells upon administration of combined dose, and it is the positive sideof this work.

Response: Yes, the combination works very well.

  1. f) The mechanistic studies of the drug-drug interaction (if-any) may be provided in this work.

Response: The mechanistic studies are in progress. They require more resources and more time. Also, it will increase the length of the manuscript. So, the mechanistic studies will be published as a new manuscript.

  1. The authors are recommended to emphasis the importance of iminosugars and sugar derivatives as an anti-diabetic agents, and it is recommended to cite following relevant articles related to iminosugars in introduction section.
  1. Nash, R. J.; Kato, A.; Yu, C-. Y.; Fleet, G. W. J. Iminosugars as therapeutic agents: recent advances and promising trends. Future Med. Chem20113, 1513−1521.
  2. Yang, L.-F.; Shimadate, Y.; Kato, A.; Li, Y.-X.; Jia, Y.-M.; Fleet, G.W.J.; Yu, C.-Y. Synthesis and glycosidase inhibition of N-substituted derivatives of DIM. Org. Biomol. Chem. 2020, 18, 999–1011.
  3. Chennaiah, A.; Dahiya, A.; Dubbu, S.; Vankar, Y. D. A Stereoselective Synthesis of an Imino Glycal: Application in the Synthesis of (−)-1-Epi -Adenophorine and a Homoiminosugar. Eur. J. Org. Chem2018, 6574−6581.
  1. Chennaiah, A.; Bhowmick, S.; Vankar, Y. D. Conversion of glycals into vicinal-1,2-diazides and 1,2-(or 2,1)-azidoacetates using hypervalent iodine reagents and Me3SiN3. Application in the synthesis of N-glycopeptides, pseudo-trisaccharides and an iminosugar. RSC Adv20177, 41755−41762.
  1. Rajasekaran, P.; Ande, C.; Vankar, Y. D. Synthesis of (5,6 & 6,6)-oxa-oxa annulated sugars as glycosidase inhibitors from 2-formyl galactal using iodocyclization as a key step. ARKIVOC 2022, vi, 5−23.

 Response: Thanks for the suggestion. We have included the references in the manuscript.

Overall, after addressing the points mentioned above, I recommend this article to publish in molecules.

Reviewer 2 Report

Summary

The present study demonstrates that puerarin and metformin have positive pharmacodynamic (PD) interactions. This manuscript is relevant to the journal's scope, scientifically sound, and well organized. The purpose of the manuscript is clearly articulated, the experimental design is appropriate, and the title accurately reflects its contents. Relevant references are provided to support the manuscript. This study contributes to a better understanding of the positive PD interactions between puerarin and metformin, and will help design more effective therapeutic approaches and clinical trials. However, there is scope for further improvement of the manuscript by incorporating the points listed below.

Major comments:

1.     The introduction or discussion section could be strengthened by providing information on the pharmacokinetics (PK) of both compounds and the pharmacokinetic PK drug interaction between them.

2.     Is this PD interaction caused by PK interaction? Discussion is required.

3.     The discussion section needs to be expanded to include specific details on the limitations of the existing study, agreement or contrast to the previously published report.

Minor comments:

1.     Throughout the text, the author should provide full names to define abbreviated terms.

2.     Verify the abbreviation whether GST or GSH?

Author Response

Summary

The present study demonstrates that puerarin and metformin have positive pharmacodynamic (PD) interactions. This manuscript is relevant to the journal's scope, scientifically sound, and well organized. The purpose of the manuscript is clearly articulated, the experimental design is appropriate, and the title accurately reflects its contents. Relevant references are provided to support the manuscript. This study contributes to a better understanding of the positive PD interactions between puerarin and metformin, and will help design more effective therapeutic approaches and clinical trials. However, there is scope for further improvement of the manuscript by incorporating the points listed below.

Major comments:

  1. The introduction or discussion section could be strengthened by providing information on the pharmacokinetics (PK) of both compounds and the pharmacokinetic PK drug interaction between them.

Response: Thanks for the suggestion. One new paragraph is introduced in the discussion section.

  1. Is this PD interaction caused by PK interaction? Discussion is required.

Response: Highly likely the PD interaction may not be caused by PK interaction, based on the reported studies. However, it is worth investigating it. A separate new paragraph is added in the section.

  1. The discussion section needs to be expanded to include specific details on the limitations of the existing study, agreement or contrast to the previously published report.

Response: The limitations of the existing study and comparisions with the pul are highlighted in the discussion section. 

Minor comments:

  1. Throughout the text, the author should provide full names to define abbreviated terms.

Response: The full names of all the abbreviated terms were provided throughout the text.

  1. Verify the abbreviation whether GST or GSH?

Response: Thanks for correcting. It should be GSH. It is changed throughout the manuscript.